# Subcellular distribution of ezrin/radixin/moesin and their roles in the cell surface localization and transport function of P-glycoprotein in human colon adenocarcinoma LS180 cells

**Takuro Kobori**, **Mayuka Tameishi, Chihiro Tanaka**, **Yoko Urashima**, **Tokio Obata** *

Laboratory of Clinical Pharmaceutics, Faculty of Pharmacy, Osaka Ohtani University, Tondabayashi, Osaka, Japan

* otbatatoki@osaka-ohtani.ac.jp

**Data Availability Statement:** All relevant data are within the manuscript.

## Abstract

The ezrin/radixin/moesin (ERM) family proteins act as linkers between the actin cytoskeleton and P-glycoprotein (P-gp) and regulate the plasma membrane localization and functionality of the latter in various cancer cells. Notably, P-gp overexpression in the plasma membrane of cancer cells is a principal factor responsible for multidrug resistance and drug-induced mutagenesis. However, it remains unknown whether the ERM proteins contribute to the plasma membrane localization and transport function of P-gp in human colorectal cancer cells in which the subcellular localization of ERM has yet to be determined. This study aimed to determine the gene expression patterns and subcellular localization of ERM and P-gp and investigate the role of ERM proteins in the plasma membrane localization and transport function of P-gp using the human colon adenocarcinoma cell line LS180. Using real-time reverse transcription polymerase chain reaction and immunofluorescence analyses, we showed higher levels of ezrin and moesin mRNAs than those of radixin mRNA in these cells and preferential distribution of all three ERM proteins on the plasma membrane. The ERM proteins were highly colocalized with P-gp. Additionally, we show that the knockdown of ezrin, but not of radixin and moesin, by RNA interference significantly decreased the cell surface expression of P-gp in LS180 cells without affecting the mRNA expression of P-gp. Furthermore, gene silencing of ezrin substantially increased the intracellular accumulation of rhodamine123, a typical P-gp substrate, with no alterations in the plasma membrane permeability of Evans blue, a passive transport marker. In conclusion, ezrin may primarily regulate the cell surface localization and transport function of P-gp as a scaffold protein without influencing the transcriptional activity of P-gp in LS180 cells. These findings should be relevant for treating colorectal cancer, which is the second leading cause of cancer-related deaths in males and females combined.

**Funding:** The authors received no specific funding for this work.

**Competing interests:** The authors have declared that no competing interests exist.

## Introduction

According to the World Health Organization, cancer is the second most common cause of death worldwide and was responsible for an estimated 9.6 million deaths in 2018 [1]. Colorectal cancer (CRC) is the second leading cause of cancer deaths among males and females combined [1]. Despite great advances in the treatment of cancer in the last few decades, effective strategies for pharmacotherapy of CRC are still under development [2–4]. One of the major problems with the treatment of CRC is the acquired multidrug resistance (MDR) of cancer cells, not only to a broad spectrum of structurally and functionally distinct conventional chemotherapeutic drugs but also to molecular targeted drugs during the course of their treatment [5–10]. Moreover, drug-induced mutagenesis is also one of the most common barriers to the clinical success for cancer chemotherapy [11, 12].

P-glycoprotein (P-gp/ABCB1/MDR1) belongs to the ATP-binding cassette (ABC) transporter superfamily, which actively exports structurally and functionally unrelated chemotherapeutic drugs from cancer cells to the extracellular space [13, 14]. Among the complex and multifactorial mechanisms underlying MDR, overexpression of P-gp in the cancer cell membrane has long been recognized as the principal factor responsible for MDR in a variety of cancers [15–18].

The members of the ezrin/radixin/moesin (ERM) family of proteins act as linkers between the actin cytoskeleton and various plasma membrane proteins, including P-gp, multidrug resistant protein-2, and epidermal growth factor receptor 2 [19–21]. Several recent studies have demonstrated that ERM play a critical role in the maintenance of cell surface membrane localization and drug efflux function of P-gp in a variety of cancer cells [17, 21–25]. However, it is largely unknown whether ERM contribute to the plasma membrane localization and drug efflux function of P-gp in human colon adenocarcinoma in which subcellular localization of ERM has yet to be determined.

The aim of this study was to identify the gene expression pattern and subcellular localization of ERM together with P-gp and to investigate the role of ERM in the plasma membrane localization and transport function of P-gp using RNA interference-mediated gene silencing experiments in LS180 cells, a representative human colon adenocarcinoma cell line.

## Materials and methods

### Cell culture

The human colon adenocarcinoma cell lines, LS180 and Caco-2, were purchased from European Collection of Cell Cultures (ECACC) collections (KAC, Hyogo, Japan). LS180 cells were cultured in Dulbecco's modified Eagle medium (DMEM) containing 1,500 mg/L glucose (FUJIFILM Wako Pure Chemical, Osaka, Japan) and Caco-2 cells were cultured in DMEM with 4,500 mg/L glucose. In both the cases, the medium was supplemented with heat-inactivated 10% fetal bovine serum (FBS) (Cosmo Bio, Tokyo, Japan). The cultures were maintained at 37˚C in a humidified atmosphere with 5% $CO_2$.

The acidic cell culture medium (pH 6.5) was obtained by the addition of 35–37% HCl solution to DMEM containing 1,500 mg/L glucose supplemented with heat-inactivated 10% FBS by measuring pH with a FiveEasy Plus FEP20 pH Meter (METTLER TOLEDO, Tokyo, Japan). LS180 cells were grown for a sufficient time in normal medium and were subsequently maintained in the acidic cell culture medium (pH 6.5) for a few days as described previously with some modifications [26–28].

## Transfection of cells with siRNAs

LS180 cells were cultured until 70–80% confluent in the normal (for all experiments) or the acidic medium (for isolation of total RNA, flow cytometric analysis, and cell viability assay), and then were seeded at a density of $2.5 \times 10^4$ cells/well in 24-well cell culture plates (Corning, Glendale, AZ, USA) for isolation of total RNA, flow cytometric analysis, rhodamine123 (Rho123) accumulation study, and passive transport assay. For Rho123 accumulation study using confocal laser microscope, $2.0 \times 10^4$ cells/dish were seeded in the center of polylysine-coated 35-mm glass bottom dish with 14-mm inner diameter (Matsunami Glass, Osaka, Japan), and for cell viability assay, $5.0 \times 10^3$ cells/well were plated in 96-well cell culture plates (Thermo Fisher Scientific, Tokyo, Japan). The cultures were incubated overnight at 37˚C in a humidified atmosphere with 5% $CO_2$ to allow for attachment. Silencer Select small interfering RNAs (siRNAs) for each target gene (Thermo Fisher Scientific) were diluted with Opti-MEM (Thermo Fisher Scientific). The cells were then transfected with 2 nM/well or 5 nM/well of siR-NAs targeting human moesin or other genes (ABCB1, ezrin, and radixin) using the Lipofecta-mine RNAiMAX Transfection Reagent (Thermo Fisher Scientific). The volume of transfection reagent used was 1.0 µL/well for total RNA isolation, Rho123 accumulation study without con-focal laser microscope, and passive transport assay, 0.8 µL/well for Rho123 accumulation study with confocal laser microscope, and 0.2 µL/well for cell viability assay. For cell viability assay, cells treated with 10 µM/well of staurosporine (Merck, Darmstadt, Germany) were included as a positive control for inducing cell death. After addition of the siRNA and transfection reagent, cells were cultured continuously for 3 days. Silencer Select Negative Control siRNA (Thermo Fisher Scientific), which has no significant similarity to human gene sequences and has mini-mal effects on gene expression, was used as a negative control for each siRNA.

## RNA isolation and real-time reverse transcription- polymerase chain reaction (RT-PCR)

Total RNA was isolated using ISOSPIN Cell & Tissue RNA (NIPPON GENE, Tokyo, Japan) according to the manufacturer's protocol. The quality and quantity of total RNA were mea-sured with a NanoDrop ND-1000 spectrophotometer (Thermo Fisher Scientific). The isolated total RNA (10–50 ng) was used as a template for real-time quantitative (RT-PCR) amplifica-tion, which was performed in 96-well optical reaction plates or Low-Profile 0.2-mL 8-Tube Strips with Optical Flat 8-Cap Strips (Bio-Rad Laboratories, Tokyo, Japan) on a CFX Connect Real-Time PCR Detection System (Bio-Rad Laboratories). The reactions were performed with a One Step TB Green Prime Script PLUS RT-PCR Kit (Takara Bio, Shiga, Japan) and the gene-specific primers for human ABCB1, ezrin, radixin, moesin, and β-actin (all purchased from Takara Bio) at a final concentration of 0.4 µM. The reaction program comprised of a reverse transcription step at 42˚C for 5 min and subsequent amplification steps including 10 s at 95˚C followed by 40 cycles of denaturation for 5 s at 95˚C and annealing for 30 s at 60˚C. The rela-tive mRNA levels of the target genes normalized to the β-actin levels (used as an internal refer-ence), amplified from the same sample were computed with the comparative quantification cycle (Cq) method ($2^{-\Delta\Delta Cq}$) using the Bio-Rad CFX Manager software version 3.1 (Bio-Rad Laboratories). The sequences of gene-specific PCR primers are given in Table 1.

## Confocal laser scanning microscopy (CLSM) analysis

CLSM analysis was conducted as described previously with some modifications [29, 30]. Briefly, LS180 cells were seeded at a density of $1.0 \times 10^5$ cells on a polylysine-coated 35-mm glass bottom dish with an inner diameter of 14-mm (Matsunami Glass) and incubated

**Table 1. Primer sequences used for RT-PCR analysis of gene expression.**

| Gene | Primer sequence (5′→3′) |
|------|------------------------|
| hβ-actin (forward) | TGGCACCCAGCACAATGAA |
| hβ-actin (reverse) | CTAAGTCATAGTCCGCCTAGAAGCA |
| hABCB1 (forward) | CACATTTGGCAAAGCTGGAGA |
| hABCB1 (reverse) | CATCATTGGCGAGCCTGGTA |
| hezrin (forward) | ACCATGGATGCAGAGCTGGAG |
| hezrin (reverse) | CATAGTGGAGGCCAAAGTACCACA |
| hradixin (forward) | GAATTTGCCATTCAGCCCAATA |
| hradixin (reverse) | GCCATGTAGAATAACCTTTGCTGTC |
| hmoesin (forward) | CCGAATCCAAGCCGTGTGTA |
| hmoesin (reverse) | GGCAAACTCCAGCTCTGCATC |

overnight 37˚C in a humidified atmosphere with 5% $CO_2$ to allow for attachment. Thereafter, the cells were washed with Dulbecco's phosphate-buffered saline (D-PBS) (Nacalai Tesque, Kyoto, Japan) and fixed with 4% paraformaldehyde (PFA) (FUJIFILM Wako Pure Chemical) for 15 min at room temperature followed by washing with D-PBS three times. Subsequently, the cells were incubated in a blocking buffer consisting of D-PBS supplemented with 10% normal goat serum (Thermo Fisher Scientific), 1% bovine serum albumin (BSA) (FUJIFILM Wako Pure Chemical), 0.3 M glycine (FUJIFILM Wako Pure Chemical), and 0.1% Tween-20 (Nacalai Tesque) for 60 min at room temperature to permeabilize the plasma membrane and to block non-specific protein–protein interactions. Thereafter, the cells were incubated overnight with a mouse anti-human P-gp antibody (Ab) (MA5-13854; Thermo Fisher Scientific) at a dilution of 1:10 in combination with a rabbit anti-human ezrin Ab (3145S; Cell Signaling Technology, Danvers, MA, USA) at a dilution of 1:50, a rabbit anti-human radixin Ab (GTX105408; GeneTex, Alton Pkwy Irvine, CA, USA) at a dilution of 1:50, or a rabbit anti-human moesin Ab (Q480) (3150S; Cell Signaling Technology) at a dilution of 1:25 in blocking buffer at 4˚C. After rinsing in D-PBS supplemented with 0.1% Tween-20 (PBS-T) for three times, the cells were incubated for 60 min at room temperature with a blocking buffer containing Alexa Fluor 488-conjugated donkey anti-mouse IgG (H+L) secondary Ab (A-21202; Thermo Fisher Scientific) at a dilution of 1:500 for P-gp or Alexa Fluor 594-conjugated goat anti-rabbit IgG (H+L) secondary Ab (R37117; Thermo Fisher Scientific) at a dilution of 1 drop/500 μL for ezrin, radixin, and moesin. The cells were then washed three times with PBS-T and sealed with a drop of Fluoro-KEEPER Antifade Reagent containing 4′,6-diamidine-2′-phenylindole dihydrochloride (DAPI) (Nacalai Tesque) for counterstaining and preserving the fluorescence. Thereafter, photomicrographs were taken at 0.3–0.4-μm intervals for *z*-axis at an original magnification of ×120 with a Nikon A1 confocal laser microscope system (Nikon Instruments, Tokyo, Japan). The two- or three-dimensional images were reconstructed from the obtained pictures using the NIS-Elements Ar Analysis software (Nikon Instruments).

## Flow cytometric analysis

Flow cytometric analysis was carried out as described previously with some modifications [29, 30]. After treatment of LS180 cells with siRNAs for 3 days, the cells were detached using 500 μL of Accutase (Nacalai Tesque) and transferred into 5 mL tubes filled with 2 mL of D-PBS and centrifuged (260 ×*g* for 5 min at 4˚C). Subsequently, the cells were incubated with a mouse anti-human Ab against P-gp (MA5-13854; Thermo Fisher Scientific) at a dilution of 0.10 μg/tube in a labeling buffer consisting of D-PBS supplemented with 5% normal horse serum (Biowest, Nuaillé, France) and 1% sodium azide (FUJIFILM Wako Pure Chemical) for

60 min at 4˚C. After rinsing in the labeling buffer and centrifugation (260 ×g for 5 min at 4˚C), the cells were incubated with Alexa Fluor 488-conjugated donkey anti-mouse IgG (H+L) secondary Ab (A-21202, Thermo Fisher Scientific) at a dilution of 1:300 in the labeling buffer for 30 min at room temperature in the dark. After rinsing in the labeling buffer and centrifugation (260 ×g for 5 min at 4˚C), the cell pellet was resuspended in 600 μL of D-PBS containing propidium iodide (PI) (Dojindo Laboratories, Kumamoto, Japan) at a concentration of 2 μg/mL to exclude PI-positive dead cells. Thereafter, the cells were analyzed with a Cell Analyzer EC800 (Sony Imaging Products & Solutions, Tokyo, Japan). Data were processed using the EC800 Analysis software (Sony Imaging Products & Solutions) to determine the mean fluorescence intensity of the Alexa Fluor 488-P-gp peak from the cell surface of LS180 cells.

## Rho123 accumulation study

The intracellular accumulation of Rho123 was determined as described previously with some modifications [22, 23, 31–34]. After treatment with siRNAs for 3 days, the LS180 cells were pretreated with 50 μM of verapamil (FUJIFILM Wako Pure Chemical), a classical inhibitor of the P-gp function [35], for 30 min followed by incubation with 50 μM of Rho123 (Merck) as a P-gp substrate in the dark for 2 h at 37˚C in a humidified atmosphere with 5% $CO_2$. Thereafter, the cells were washed three times with D-PBS to remove the residual Rho123 and lysed in 300 μL of D-PBS containing 0.1% sodium dodecyl sulfate (SDS) (Nacalai Tesque) for 30 min on ice with vigorous vortexing followed by centrifugation (15.700 ×g for 10 min at 4˚C) to extract the total protein. The amount of total cellular proteins and Rho123 concentrations in the cell lysate were quantified by measuring the absorbance at 562 nm using a Bicinchoninic acid Protein Assay Kit with BSA as a standard (Takara Bio), and the fluorescence intensity (excitation at 485 nm and emission at 538 nm), respectively with a Synergy HTX Multi-Mode Microplate Reader (BioTek Instrument, Winooski, VT, USA). The intracellular accumulation rate of Rho123 normalized to the total protein content (pmol/μg protein) was calculated by dividing the intracellular concentration of Rho123 (pmol/mL) by the total protein amount (μg/mL).

To determine the retention level of Rho123 in LS180 cells by CLSM analysis, cells were subjected to the same procedure except for the final concentration of Rho123 (10 μM) as described above before their lysis in 0.1% SDS solution. Thereafter, photomicrographs were taken at 1.0-μm intervals for z-axis at an original magnification of 20× with a Nikon A1 confocal laser microscope system (Nikon Instruments). The two- or three-dimensional images were reconstructed from the obtained pictures using the NIS-Elements Ar Analysis software (Nikon Instruments).

## Passive transport assay

After treatment with siRNAs for 3 days, the cells were pretreated for 30 min in the presence or absence of 70% ethanol (Nacalai Tesque) diluted with the complete growth medium as a positive control followed by incubation with 25 mM of Evans blue, a passive transport marker [22, 36], for 30 min at 37˚C in a humidified atmosphere with 5% $CO_2$. After removing the residual Evans blue from each well, the intracellular accumulation of dye was observed using the Eclipse Ti2 inverted microscope system (Nikon Instruments).

## Cell viability assay

After treatment of cells with siRNAs or staurosporine for 3 days, 10 μL/well of a commercial PrestoBlue Cell Viability Reagent (Thermo Fisher Scientific) was added directly to the wells containing 100 μL of the complete growth medium and the cells were incubated for 10 min at

37˚C in a humidified atmosphere with 5% $CO_2$ protected from direct light. Thereafter, fluorescence signals were detected at a wavelength of 560 nm (excitation) and 590 nm (emission) using a Synergy HTX Multi-Mode Microplate Reader (BioTek Instrument, Winooski, VT, USA). PrestoBlue is a new resazurin-based reagent to assess cell viability and cytotoxicity with higher sensitivity than 3-(4,5-dimethyl-2-thiazolyl)-2,5-diphenyl-2H-tetrazolium bromide (MTT), and comparable with that of Alamar Blue [37–40].

## Statistical analysis

Data are presented as means ± standard error of the mean (SEM). Statistical analysis was performed using the Prism version 3 software (GraphPad Software, La Jolla, CA, USA). Statistical significance was assessed by a one-way analysis of variance (ANOVA) followed by Tukey's test for multiple comparisons. Differences with $P$ values < 0.05 were considered significant.

## Results

### ERM expression profiles and localization in LS180 cells

We first measured the expression of ezrin, radixin, and moesin mRNAs in LS180 cells along with that of ABCB1 by real-time RT-PCR analysis. The expression of ezrin, moesin, and ABCB1 was high and that of radixin was low (Fig 1A). By contrast, higher levels of ezrin, radixin, and ABCB1 and lower levels of moesin mRNAs were observed in Caco-2 cells (Fig 1B). Next, subcellular localization of ERM in LS180 cells was assessed by immunofluorescence staining. CLSM analysis revealed that ezrin, radixin, and moesin (red) were preferentially distributed on the plasma membrane and colocalized with P-gp (green) (Fig 1C).

### Effect of siRNAs against each ERM on the expression levels of target mRNAs in LS180 cells

To confirm the efficiency of siRNAs against each ERM, the mRNA levels of each ERM were measured 3 days after transfection of LS180 cells with the siRNAs. siRNAs for ezrin, radixin, and moesin achieved 85%–90% reduction in the expression of their target mRNAs compared with transfection reagent alone without any influence on the expression of other genes (Fig 2A–2C). Notably, none of the siRNAs used in this study altered the viability of LS180 cells (Fig 2D).

### Effect of ERM silencing on the expression of P-gp in LS180 cells

First, we checked the effect of siRNAs against ezrin, radixin, and moesin on the level of ABCB1 mRNA. None of these siRNAs had any effect on the levels of ABCB1 mRNA, whereas the siRNA against ABCB1 significantly decreased the levels of ABCB1 mRNA, compared to that in cells treated with the transfection reagent alone (Fig 3A). Next, we investigated whether gene silencing of ezrin, radixin, and moesin affects the expression of P-gp on the surface of the plasma membrane. The results of flow cytometric analysis showed that silencing of ezrin, but not of radixin and moesin, significantly decreased the expression of P-gp on the surface of the plasma membrane to the same level as did the silencing of ABCB1 (Fig 3B and 3C).

### Effect of siRNAs against each ERM on the expression levels of target mRNAs in LS180 cells under the acidic condition

To confirm whether siRNAs against each ERM have similar effects on the target mRNA in the acidic microenvironmental condition, mRNA levels of each ERM were measured 3 days after

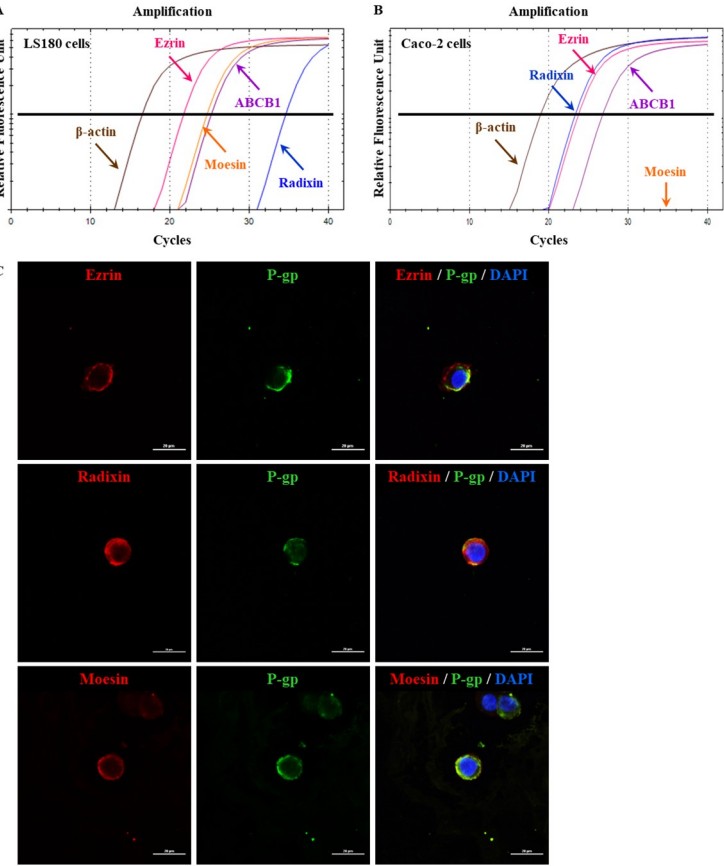

**Fig 1. Gene expression pattern and subcellular localization of ezrin, radixin, and moesin (ERM) and P-gp in LS180 cells.** (A, B) Representative amplification curves of ezrin, radixin, moesin, and ABCB1 together with that of β-actin (internal control) in LS180 and Caco-2 cells as determined by real-time quantitative reverse transcription-polymerase chain reaction. (C) Confocal laser scanning microscopy analysis of ERM and P-gp to visualize the subcellular distribution in LS180 cells. In a three-dimensional reconstruction of optically sectioned LS180 cells, ezrin, radixin, and moesin (red) were detected on the plasma membrane and preferentially colocalized with P-gp (green) on the plasma membrane. Scale bars: 20 μm. All data are representative of at least three independent experiments.

transfection of LS180 cells with the siRNAs under the acidic condition with pH 6.5. siRNAs for ezrin, radixin, and moesin achieved 74%–85% reduction in the expression of their target mRNAs compared with transfection reagent alone without any significant influence on the expression of other genes (Fig 4A–4C). Notably, none of the siRNAs used in this study also altered the viability of LS180 cells under the acidic condition with pH 6.5 (Fig 4D).

## Effect of ERM silencing on the expression of P-gp in LS180 cells under the acidic condition

We asked whether the gene silencing of ezrin, radixin, and moesin influence the level of ABCB1 mRNA and the expression of P-gp on the surface of the plasma membrane in LS180 cells under the acidic condition. None of these siRNAs had any effect on the levels of ABCB1 mRNA, whereas the siRNA against ABCB1 significantly decreased the levels of ABCB1 mRNA, compared to that in cells treated with the transfection reagent alone (Fig 5A). The results of flow cytometric analysis showed that silencing of ezrin, but not of radixin and

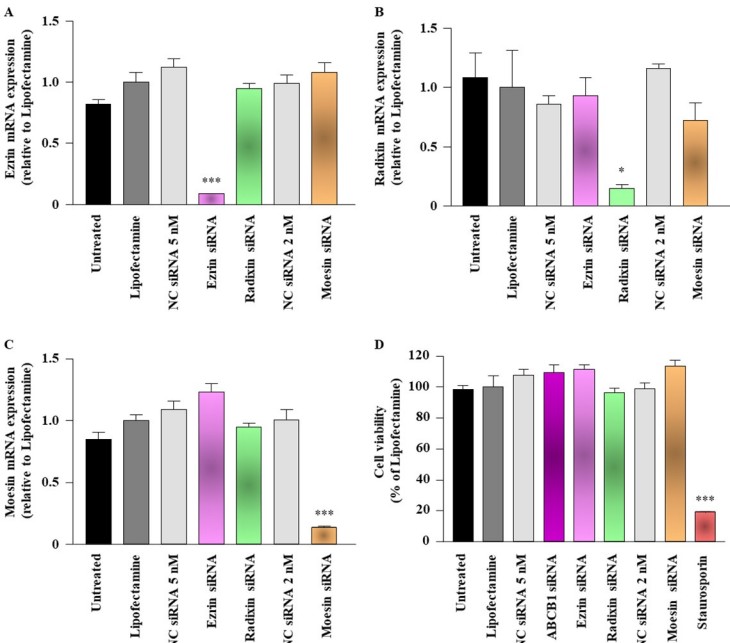

**Fig 2. Effect of siRNAs targeting ezrin, radixin, or moesin on the mRNA expression of ezrin, radixin, and moesin (ERM) in LS180 cells.** Cells were treated with the transfection medium (Untreated), transfection reagent (Lipofectamine), nontargeting control (NC) siRNA, and specific siRNAs for ezrin, radixin, or moesin and then incubated for 3 days. (A–C) Expression of each mRNA in the cells treated with siRNAs relative to that in cells treated with the transfection reagent alone was measured by real-time quantitative reverse transcription-polymerase chain reaction. n = 3, $^{***}p < 0.001$, $^{*}p < 0.05$ vs. Lipofectamine. (D) Cell viability was assessed with the PrestoBlue cell viability reagent. Staurosporine was used as a positive control for inducing cell death. n = 4, $^{***}p < 0.001$ vs. Lipofectamine. All data are expressed as the mean ± SEM and were analyzed by one-way ANOVA followed by Tukey's test.

moesin, significantly decreased the expression of P-gp on the surface of the plasma membrane to the same level as did the silencing of ABCB1 (Fig 5B and 5C).

## Effect of ERM silencing on the transport function of P-gp in LS180 cells

To evaluate the effect of ERM knockdown on the transport function of P-gp, we assessed the intracellular accumulation of Rho123 dye in LS180 cells treated with siRNAs targeting ezrin, radixin, or moesin for 3 days. The siRNA-mediated silencing of P-gp and inhibition of the transport function of P-gp by pretreatment with 50 μM of verapamil resulted in approximately 1.6-fold increase in the intracellular accumulation of Rho123 relative to that in the treatment with the transfection reagent alone. Similarly, transfection of cells with the siRNA against ezrin resulted in approximately 1.6-fold higher intracellular accumulation of Rho123 than in cells treated with the transfection reagent alone. By contrast, we observed no alterations in the rate of Rho123 uptake in LS180 cells transfected with siRNAs against radixin and moesin (Fig 6A). Similarly, the uptake capacity of Rho123 was obviously enhanced in cells transfected with siRNAs targeting ezrin and ABCB1 but not with those targeting radixin and moesin, as was also observed in cells pretreated with 50 μM of verapamil (Fig 6B). Of note, we validated that treatment of LS180 cell with 5, 10, and 50 μM of verapamil dose-dependently increased the uptake capacity of Rho123 (S1 Fig).

To exclude the possibility that the intracellular accumulation of Rho123 after treatment of LS180 cells with siRNAs was due to a loss of the plasma membrane integrity, we checked the

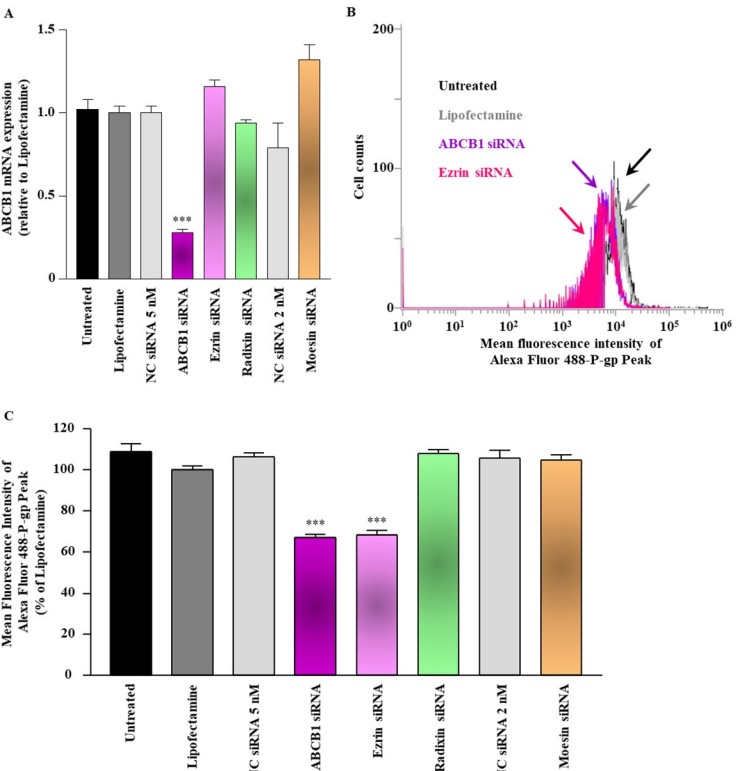

**Fig 3. Effect of siRNAs targeting ezrin, radixin, or moesin on the ABCB1 mRNA and P-gp expression on the surface of LS180 cells.** Cells were treated with the transfection medium (Untreated), transfection reagent (Lipofectamine), nontargeting control (NC) siRNA, and specific siRNAs for ABCB1, ezrin, radixin, or moesin and then incubated for 3 days. (A) The expression of ABCB1 mRNA in cells treated with each siRNA relative to that in cells treated with the transfection reagent alone was determined by real-time quantitative reverse transcription-polymerase chain reaction. n = 3, ***$p < 0.001$ vs. Lipofectamine. (B) An overlay of the representative histograms for the expression peak of Alexa Fluor 488-labeled P-gp on the surface of the plasma membrane of LS180 cells treated with the transfection medium (Untreated; black line), transfection reagent (Lipofectamine; gray line), ABCB1 siRNA (purple line), and ezrin siRNA (pink line), as measured by flow cytometry. The calculated relative mean fluorescence intensity peaks of P-gp on the surface of the plasma membrane for all the treatments are shown in (C). n = 4, ***$p < 0.001$ vs. Lipofectamine. All data are expressed as the mean ± SEM and were analyzed by a one-way ANOVA followed by Tukey's test.

membrane permeability of LS180 cells treated with each siRNA to Evans blue dye, which is absorbed by passive diffusion [22, 36]. As observed for the transfection reagent alone and negative control siRNA, siRNAs for each target gene had no influence on the intracellular accumulation of Evans blue in the presence of 70% ethanol, which was used as a positive control to disrupt the membrane integrity [41] and apparently enhanced the cellular uptake of Evans blue (Fig 7).

## Discussion

In the present study, we determined the gene expression profiles of ezrin, radixin, and moesin together with that of P-gp in LS180 cells, and found them to be different from those in Caco-2 cells. Interestingly, our CLSM analysis data demonstrate for the first time that ERM are preferentially localized to the plasma membrane and are highly colocalized with P-gp. Nowak et al. indicated that in various kinds of human colon adenocarcinoma including LS180 cells, ezrin is distributed at the tips of pseudopodia, localized as a sharp ring under the cellular membrane,

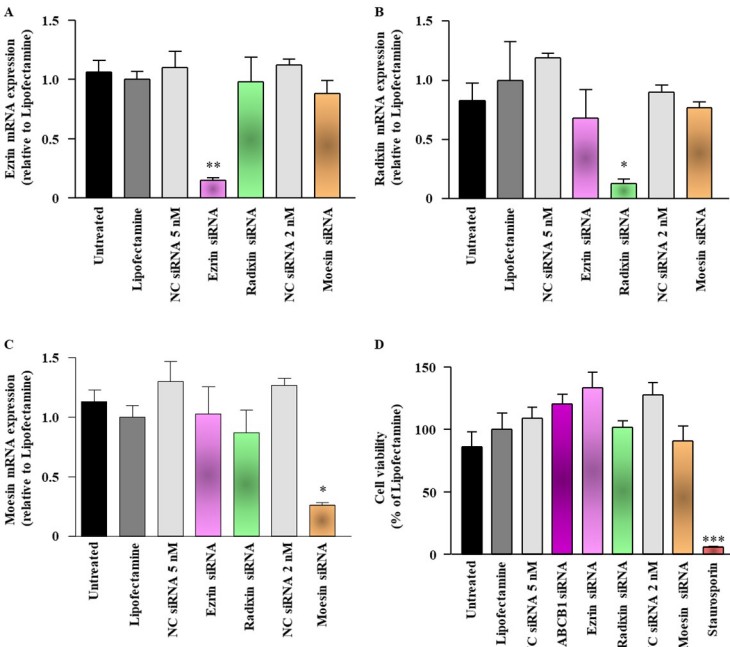

**Fig 4. Effect of siRNAs targeting ezrin, radixin, or moesin on the mRNA expression of ezrin, radixin, and moesin (ERM) in LS180 cells under the acidic condition.** Cells were treated with the transfection medium (Untreated), transfection reagent (Lipofectamine), nontargeting control (NC) siRNA, and specific siRNAs for ezrin, radixin, or moesin and then incubated for 3 days under the acidic condition with pH 6.5. (A–C) Expression of each mRNA in the cells treated with siRNAs relative to that in cells treated with the transfection reagent alone was measured by real-time quantitative reverse transcription-polymerase chain reaction. n = 3, $^{**}p < 0.01$, $^{*}p < 0.05$ vs. Lipofectamine. (D) Cell viability was assessed with the PrestoBlue cell viability reagent. Staurosporine was used as a positive control for inducing cell death. n = 6, $^{***}p < 0.001$ vs. Lipofectamine. All data are expressed as the mean ± SEM and were analyzed by one-way ANOVA followed by Tukey's test.

or is present at the edges in areas of subtle protrusions [42]. Moreover, abundant levels of radixin were detected in human colon tumor tissue [43] and also in a variety of human colon cancer cell lines [44]. Unlike LS180 cells, Caco-2 cells were deficient in moesin, which is in agreement with previous findings [45], implying a characteristic feature of LS180 cells, which carries genes for all three ERM proteins. Accumulating evidence indicates that P-gp is highly expressed not only in the cells derived from human gastrointestinal epithelium, including LS180 and Caco-2 cells [46–48], but also in the gastrointestinal tissues of rodents [49–54]. These previous findings strongly support the results obtained in the present study, indicating the gene expression pattern and subcellular localization of ERM along with P-gp in LS180 cells.

To verify the role of ERM in the expression and transport activity of P-gp, we used siRNAs to silence ezrin, radixin, and moesin in LS180 cells. We validated that each siRNA targeting the ERM proteins substantially and specifically reduced the level of respective target mRNAs without causing any cytotoxicity in LS180 cells. Therefore, we succeeded in developing an *in vitro* experimental model to determine the role of ERM in the expression and transport function of P-gp in LS180 cells.

A growing body of evidence suggests that the protein expression level and functional activity of transporters, such as P-gp, are not always dependent on their mRNA expression levels [55–57]. As localization to the plasma membrane is required for the functionality of P-gp and other transporters [13, 20, 57–61], scaffold proteins, which anchor the transporters to the

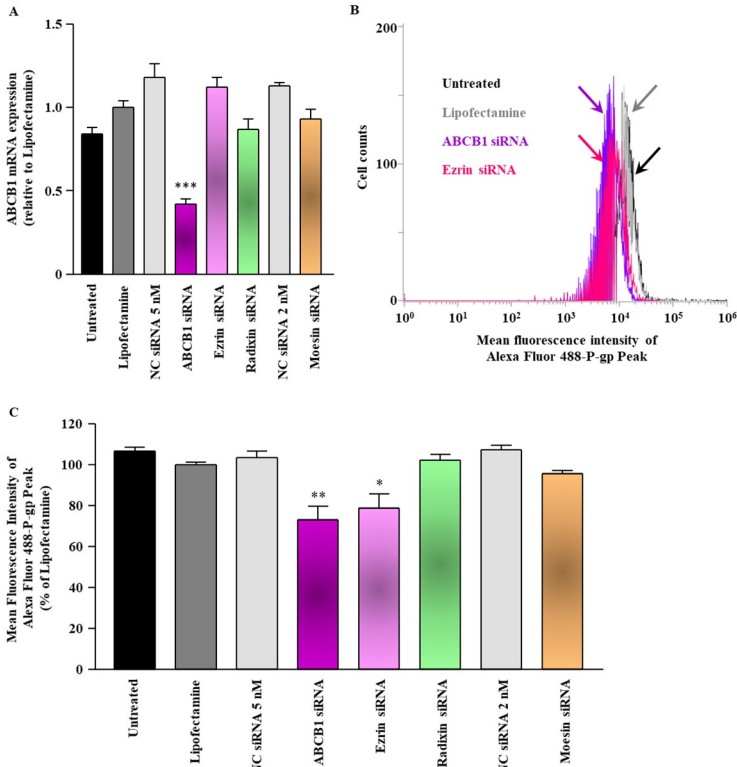

**Fig 5. Effect of siRNAs targeting ezrin, radixin, or moesin on the ABCB1 mRNA and P-gp expression on the surface of LS180 cells under the acidic condition.** Cells were treated with the transfection medium (Untreated), transfection reagent (Lipofectamine), nontargeting control (NC) siRNA, and specific siRNAs for ABCB1, ezrin, radixin, or moesin and then incubated for 3 days under the acidic condition with pH 6.5. (A) The expression of ABCB1 mRNA in cells treated with each siRNA relative to that in cells treated with the transfection reagent alone was determined by real-time quantitative reverse transcription-polymerase chain reaction. n = 3, $^{***}p < 0.001$ vs. Lipofectamine. (B) An overlay of the representative histograms for the expression peak of Alexa Fluor 488-labeled P-gp on the surface of the plasma membrane of LS180 cells treated with the transfection medium (Untreated; black line), transfection reagent (Lipofectamine; gray line), ABCB1 siRNA (purple line), and ezrin siRNA (pink line), as measured by flow cytometry. The calculated relative mean fluorescence intensity peaks of P-gp on the surface of the plasma membrane for all the treatments are shown in (C). n = 4, $^{**}p < 0.01$, $^{*}p < 0.05$ vs. Lipofectamine. All data are expressed as the mean ± SEM and were analyzed by a one-way ANOVA followed by Tukey's test.

surface of the plasma membrane, have been recognized as critical regulators of several transporters by post-translational modification. Notably, the ERM proteins are closely related with the localization of P-gp to the surface of the plasma membrane and with its functional activity in various kinds of tissues and cells [17, 21–25, 62–64]. In the present study, we performed gene-silencing experiments using siRNAs to investigate whether the ERM proteins functionally regulate P-gp in LS180 cells. Interestingly, siRNA against ezrin substantially decreased the cell surface expression of P-gp without affecting the levels of the ABCB1 mRNA. On the contrary, siRNAs against radixin and moesin did not cause any alterations in the levels of ABCB1 mRNA or its cell surface localization. Kobori et al. has previously shown that repeated oral administration of etoposide, an anticancer drug and a strong substrate of P-gp, to mice increases the expression of P-gp via the activation of radixin, in the plasma membrane of small intestine where the protein–protein interaction between P-gp and radixin was detected by immunoprecipitation analysis [51, 65–67]. Of note, not only radixin but also ezrin and moesin interacted with P-gp in the plasma membrane fraction of the small intestine in mice [65, 66].

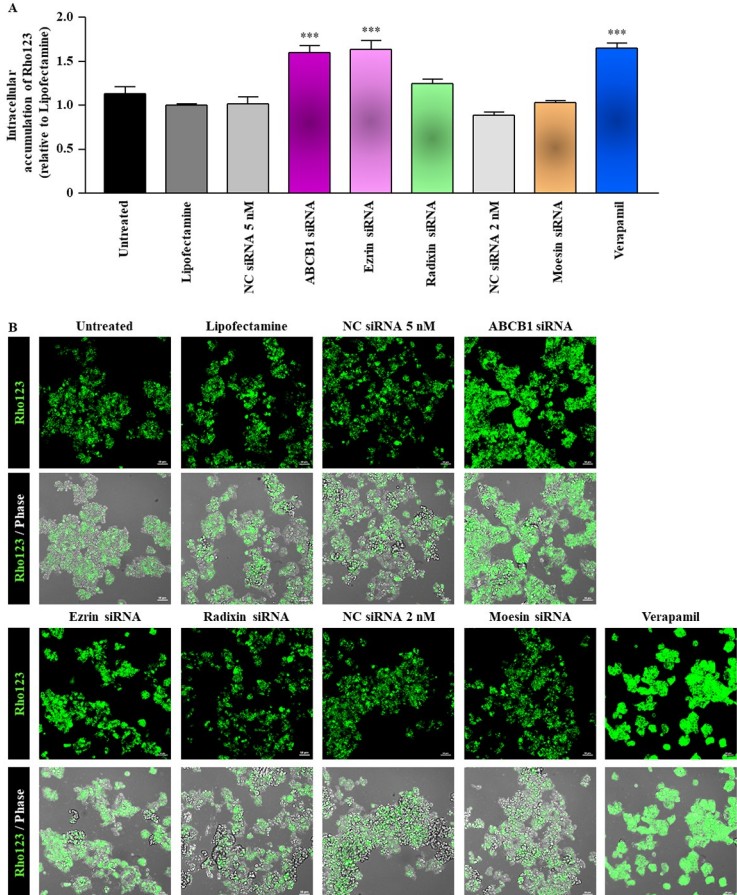

**Fig 6. Intracellular accumulation of Rho123 in LS180 cells after treatment with siRNA.** Cells were treated with the transfection medium (Untreated), transfection reagent (Lipofectamine), nontargeting control (NC) siRNA, and specific siRNAs for ABCB1, ezrin, radixin, or moesin and then incubated for 3 days. (A) The intracellular accumulation rate of Rhodamine (Rho) 123 normalized to the total protein amount (pmol/μg protein) relative to the accumulation in cells treated with the transfection reagent alone. n = 6, *** $p < 0.001$ vs. Lipofectamine. Data are expressed as the mean ± SEM and were analyzed by one-way ANOVA followed by Tukey's test. (B) Representative confocal laser scanning microscopy images of fluorescent Rho123 and of each overlay with phase contrast in LS180 cells. Upper and lower panels indicate Rho123 and merge of Rho123 with phase contrast image, respectively. Scale bars: 50 μm. All the images are representative of at least three independent experiments.

Intriguingly, Jin et al. showed that the overexpression of sphingomyelin synthase 1 in Caco-2 cells substantially increases the protein expression of ezrin along with an enhancement in the protein expression and functional activity of P-gp [32]. These findings raise the possibility of a close relationship between ezrin and P-gp mediated by the potential protein–protein interaction in the plasma membrane of LS180 cells. Together these results suggest that ezrin may solely regulate the localization of P-gp to the surface of the plasma membrane, possibly by post-translational modification as a scaffold protein without influencing the regulatory mechanism for the transcription of the ABCB1 gene in LS180 cells.

Another importance is that gene silencing of ezrin, but not of radixin and moesin significantly decreased the cell surface expression of P-gp without affecting the levels of the ABCB1 mRNA in LS180 cells cultured at the acidic pH condition which mimics the actual tumor microenvironment. These results are in good agreement with those observed in the normal culture condition. The extracellular acidity is a common hallmark of almost all tumors due to

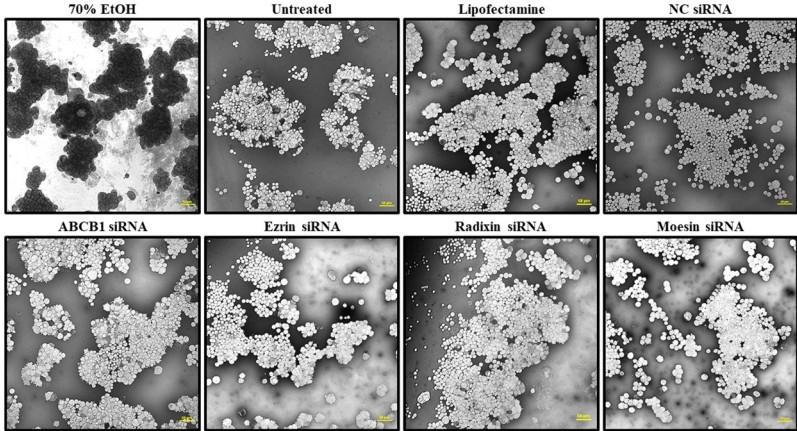

**Fig 7. Effect of siRNA treatment on passive transport in LS180 cells.** The phase-contrast images of Evans blue accumulation in LS180 cells pretreated with 70% ethanol (EtOH) to disrupt the plasma membrane integrity and of those treated with siRNAs targeting ABCB1, ezrin, radixin, or moesin for 3 days. Scale bars: 50 μm. All the images are representative of at least three independent experiments.

the accumulation of lactic acid and H⁺, leading to the microenvironmental acidification [28, 68, 69]. Previous studies have shown that microenvironmental acidic pH exerts divergent roles in tumor development such as invasion, dissemination, and metastasis [70–72]. Moreover, a growing body of recent evidence suggests that acquired MDR not only to a conventional chemotherapeutics but also to molecular targeted drugs can also be caused by the highly acidic microenvironment of tumors [69]. Taken together, ezrin may play a key role in the plasma membrane localization of P-gp at the acidic tumor microenvironment, leading to an acquired MDR in a variety of cancer cells.

In addition, accumulated evidence suggests that ezrin plays an essential role in both the maintenance of migratory and invasive capacity of tumor cells through the specific interaction with some tumor-associated plasma membrane proteins, which leads to the metastatic behavior of tumor cells [73–75]. Therefore, ezrin may exert multiple function not only in driving the MDR but also in acquiring metastatic phenotype of cancer cells in human tumors.

We also observed that the knockdown of ezrin gene, but not of radixin and moesin gene, markedly increased the intracellular accumulation of Rho123 in LS180 cells. On the contrary, no changes in the cellular uptake of Evans blue, a passive diffusion marker, were found in cells treated with siRNAs against each of the ERM proteins or P-gp, implying that an increase in the uptake rate of Rho123 induced by the silencing of ezrin gene could not arise from a disruption of membrane integrity in LS180 cells. By contrast, Yano et al. demonstrated a significant increase in the intracellular accumulation of Rho123 in Caco-2 cells upon siRNA-mediated silencing of radixin gene and a substantial decrease in the Rho123 efflux rate in the small intestine of radixin-deficient mice, indicating a reduction in the transport activity of P-gp by the knockdown and knockout of the radixin gene [22, 23, 52]. This discrepancy between the results obtained in the present and previous studies may be attributed at least in part to the different expression profile of ERM in these cell lines. In fact, as evident from the results of the present and previous studies [45, 76], the ezrin mRNA was abundantly expressed not only in LS180 cells but also in Caco-2 cells, whereas higher or lower expression of radixin was detected in Caco-2 and LS180 cells, respectively, which may lead to the variation in the scaffold protein regulating the plasma membrane localization and transport function of P-gp, although the precise details are yet to be elucidated. Taken together, ezrin might serve as an anchor protein to

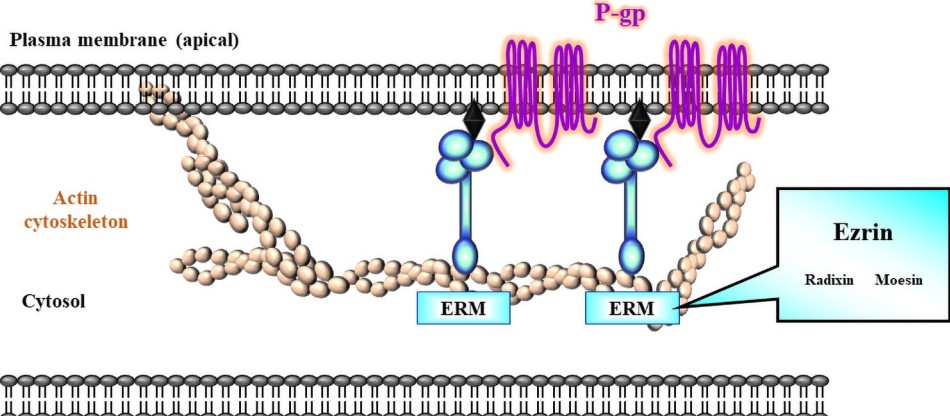

**Fig 8. Schematic diagram of the proposed mechanism by which ezrin contributes to the cell surface localization and functionality of P-gp in LS180 cells.** Among the ERM proteins, ezrin plays a key role as an anchor protein to stabilize P-gp on the surface of the plasma membrane, which in turn regulates the transport function of P-gp in LS180 cells.

immobilize P-gp on the surface of the plasma membrane, which in turn maintains the transport function of P-gp in LS180 cells (Fig 8).

## Conclusions

We demonstrate that ezrin and moesin mRNAs are abundantly expressed in LS180 cells whereas the expression of radixin mRNA is weak. Moreover, all three ERM proteins are specifically localized on the plasma membrane of LS180 cells where they are highly colocalized with P-gp. Notably, ezrin primarily regulates the cell surface localization and functionality of P-gp possibly via its post-translational modification as a scaffold protein without influencing the transcriptional level of P-gp in LS180 cells. These results raise the possibility that specific suppression of ezrin expression might increase the intracellular accumulation of P-gp substrate drugs, such as anti-cancer agents, in cancer cells, which may in turn lead to an improvement in the pharmacotherapeutic efficacy of these drugs.

## Supporting information

**S1 Fig. Intracellular accumulation of rhodamine123 in LS180 cells after treatment with verapamil in a concentration-dependent manner.**
(PDF)

## Acknowledgments

The authors would like to thank Editage (https://www.editage.jp/) for their assistance with English language editing.

## Author Contributions

**Conceptualization:** Takuro Kobori.

**Data curation:** Takuro Kobori.

**Formal analysis:** Takuro Kobori.

**Investigation:** Takuro Kobori, Mayuka Tameishi, Chihiro Tanaka.

**Methodology:** Takuro Kobori, Chihiro Tanaka, Yoko Urashima.

**Project administration:** Takuro Kobori, Yoko Urashima, Tokio Obata.

**Supervision:** Tokio Obata.

**Validation:** Mayuka Tameishi.

**Visualization:** Mayuka Tameishi.

**Writing – original draft:** Takuro Kobori.

**Writing – review & editing:** Takuro Kobori, Yoko Urashima, Tokio Obata.

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
