## [Decision Letter · Decision Letter 0]

15 Mar 2021

PONE-D-21-04413

Subcellular distribution of ezrin/radixin/moesin and their roles in the cell surface localization and transport function of P-glycoprotein in human colon adenocarcinoma LS180 cells

PLOS ONE

Dear Dr. Obata,

Thank you for submitting your manuscript to PLOS ONE. After careful consideration, we feel that it has merit but does not fully meet PLOS ONE’s publication criteria as it currently stands. Therefore, we invite you to submit a revised version of the manuscript that addresses the points raised during the review process.

Both reviewers have identified a number of issues which should be addressed if the authors plan to submit a revised manuscript. 

We look forward to receiving your revised manuscript.

Kind regards,

Salvatore V Pizzo

Academic Editor

PLOS ONE

Journal Requirements:

Reviewers' comments:

Reviewer's Responses to Questions

**Comments to the Author**

1. Is the manuscript technically sound, and do the data support the conclusions?

Reviewer #1: Partly

Reviewer #2: Yes

2. Has the statistical analysis been performed appropriately and rigorously? 

Reviewer #1: N/A

Reviewer #2: Yes

3. Have the authors made all data underlying the findings in their manuscript fully available?

Reviewer #1: Yes

Reviewer #2: Yes

4. Is the manuscript presented in an intelligible fashion and written in standard English?

Reviewer #1: Yes

Reviewer #2: Yes

5. Review Comments to the Author

Reviewer #1: Subcellular d 1 istribution of ezrin/radixin/moesin and their roles in the cell surface localization and transport function of P-glycoprotein in human colon adenocarcinoma LS180 cells

By Takuro Kobori, Yoko Urashima, Tokio Obata

This reviewer is happy to read about the role of ERM proteins in P-gp participation to human tumor cells’ drug resistance.

The issue was really warm at the beginning of 2000, with some papers showing the importance of membrane to cytoskeleton connection in P-gp-mediated drug resistance. This seemed directly related to the importance of the membrane-skeleton apparatus in allowing cellular polarization in driving many cellular funzions in both normal conditions (Fais S et al Immunology Today 1997; Parlato S et al EMBO J 2000; Luciani F et al Cell Death Diff 2004; Ramoni C et al Eur J Immunol 2002; Fais S et al J Leukoc Biol. 2003), but also in promoting the metastatic behaviour of tumor cells (Fais S Leuk Res. 2010) and the cell-in-cell phenomena (reviewed in Fais S, Overholtzer M, Nature Reviews Cancer 2018). I guess the authors should generally discuss better the importance of ERM-mediated mebrane-to-skeleton connection in both normal and tumor condition.

As far as the experimental protocol is concerned, the authors should add more to the issue, that can’t be limited to the use of human colon cancer cell, inasmuc as this is a mechanism that involves virtually all cancer histotypes. One suggestion might be to consider that there is another mechanism of tumor drug resistance that while very simple per se is very efficient in rendering cancer cells resistant to chemotherapeutics actually, that is microenvironmental acidity. It has been shown that it is a very powerfull mechanism of resistance (reviewed in Pillai et al Cancer Metastasis Rev. 2019), that may overcome P-gp-mediated drug resistance as well (Luciani F et al J Natl Cancer Inst. 2004). There are models of the same cell line cultured at either the standard 7,4 pH condition or at the acidic pH of tumors (i.e. 6,5), and this includes human colon cancer as well (Logozzi M et al Cancers 2018).

The authors should test the involvement of both P-gp and the ERM-mediated membrane-to-cytoskeleton connection at different pH, in order to render their findings more real.

Reviewer #2: The current study elucidated the interaction of ERM and P-gp on the membrane of LS180 cells, and the authors found that ezrin but ont radixin and moesin closely regulated P-pg function parabably via protein-protein interaction. Given that P-gp inhibitors have encountered many failures, this study may provide an alternative to suppress MDR by targeting ezrin. Overall, the methods and statistical analysis are appropriate, and the writing are good. This is a well-designed and performed study. It's my opinion that it can be published after minor revison.

1, line 233, the the

2, The Figures are in low resolution that were hard to identify.

3, line 301, would it be too high concentrition of 50 μM of verapamil?

4, ABCB1 mRNA vs. P-gp mRNA, they are inconsistant in Results and Discussion (line 373)

6. PLOS authors have the option to publish the peer review history of their article (what does this mean?). If published, this will include your full peer review and any attached files.

Reviewer #1: No

Reviewer #2: No

---

## [Author Response · Author response to Decision Letter 0]

31 Mar 2021

Reply comments to #Reviewer 1

We would like to thank #Reviewer 1 for valuable comments and suggestions on our manuscript. We have carefully read all of your comments and suggestions and have made the corrections in the revised version of manuscript. Detailed responses to your comments are listed below, and we highlighted all changes with word track changes in the file labeled ‘Revised Manuscript with Track Changes’. We hope this revised manuscript would be satisfactory for publication in PLoS One.

Comment 1. The authors should test the involvement of both P-gp and the ERM-mediated membrane-to-cytoskeleton connection at different pH, in order to render their findings more real.

Reply Comments.

We would like to appreciate #Reviewer 1’s valuable suggestion. As #Reviewer 1 pointed out, we performed additional study to investigate whether ERM proteins also regulate the plasma membrane localization of P-gp under the condition at the acidic pH of tumor microenvironment. Based on the results of additional experiments, we added following sentence in the Materials and Methods, Results, Discussion, and References to the revised version of manuscript.

Materials and Methods (Line 68 – 73)

The acidic cell culture medium (pH 6.5) was obtained by the addition of 35–37% HCl solution to DMEM containing 1,500 mg/L glucose supplemented with heat-inactivated 10% FBS by measuring pH with a FiveEasy Plus FEP20 pH Meter (METTLER TOLEDO, Tokyo, Japan). LS180 cells were grown for a sufficient time in normal medium and were subsequently maintained in the acidic cell culture medium (pH 6.5) for a few days as described previously with some modifications [26-28].

Materials and Methods (Line 76 – 78)

LS180 cells were cultured until 70–80% confluent in the normal (for all experiments) or the acidic medium (for isolation of total RNA, flow cytometric analysis, and cell viability assay), and then ～.

Results (Line 304 – 355)

Effect of siRNAs against each ERM on the expression levels of target mRNAs in LS180 cells under the acidic condition.

 To confirm whether siRNAs against each ERM have similar effects on the target mRNA in the acidic microenvironmental condition, mRNA levels of each ERM were measured 3 days after transfection of LS180 cells with the siRNAs under the acidic condition with pH 6.5. siRNAs for ezrin, radixin, and moesin achieved 74%–85% reduction in the expression of their target mRNAs compared with transfection reagent alone without any significant influence on the expression of other genes (Figs 4A–C). Notably, none of the siRNAs used in this study also altered the viability of LS180 cells under the acidic condition with pH 6.5 (Fig 4D).

Fig 4. Effect of siRNAs Targeting Ezrin, Radixin, or Moesin on the mRNA Expression of Ezrin, Radixin, and Moesin (ERM) in LS180 cells under the Acidic Condition. Cells were treated with the transfection medium (Untreated), transfection reagent (Lipofectamine), nontargeting control (NC) siRNA, and specific siRNAs for ezrin, radixin, or moesin and then incubated for 3 days under the acidic condition with pH 6.5. (A–C) Expression of each mRNA in the cells treated with siRNAs relative to that in cells treated with the transfection reagent alone was measured by real-time quantitative reverse transcription-polymerase chain reaction. n = 3, **p < 0.01, *p < 0.05 vs. Lipofectamine. (D) Cell viability was assessed with the PrestoBlue cell viability reagent. Staurosporine was used as a positive control for inducing cell death. n = 6, ***p < 0.001 vs. Lipofectamine. All data are expressed as the mean ± SEM and were analyzed by one-way ANOVA followed by Tukey’s test. 

Effect of ERM Silencing on the Expression of P-gp in LS180 Cells under the acidic condition.

 We asked whether the gene silencing of ezrin, radixin, and moesin influence the level of ABCB1 mRNA and the expression of P-gp on the surface of the plasma membrane in LS180 cells under the acidic condition. None of these siRNAs had any effect on the levels of ABCB1 mRNA, whereas the siRNA against ABCB1 significantly decreased the levels of ABCB1 mRNA, compared to that in cells treated with the transfection reagent alone (Fig 5A). The results of flow cytometric analysis showed that silencing of ezrin, but not of radixin and moesin, significantly decreased the expression of P-gp on the surface of the plasma membrane to the same level as did the silencing of ABCB1 (Figs 5B, C).

Fig 5. Effect of siRNAs Targeting Ezrin, Radixin, or Moesin on the ABCB1 mRNA and P-gp Expression on the Surface of LS180 cells under the Acidic Condition. Cells were treated with the transfection medium (Untreated), transfection reagent (Lipofectamine), nontargeting control (NC) siRNA, and specific siRNAs for ABCB1, ezrin, radixin, or moesin and then incubated for 3 days under the acidic condition with pH 6.5. (A) The expression of ABCB1 mRNA in cells treated with each siRNA relative to that in cells treated with the transfection reagent alone was determined by real-time quantitative reverse transcription-polymerase chain reaction. n = 3, ***p < 0.001 vs. Lipofectamine. (B) An overlay of the representative histograms for the expression peak of Alexa Fluor 488-labeled P-gp on the surface of the plasma membrane of LS180 cells treated with the transfection medium (Untreated; black line), transfection reagent (Lipofectamine; gray line), ABCB1 siRNA (purple line), and ezrin siRNA (pink line), as measured by flow cytometry. The calculated relative mean fluorescence intensity peaks of P-gp on the surface of the plasma membrane for all the treatments are shown in (C). n = 4, **p < 0.01, *p < 0.05 vs. Lipofectamine. All data are expressed as the mean ± SEM and were analyzed by a one-way ANOVA followed by Tukey’s test.

Figure 4 and 5 (See attached file named Fig 4 and Fig 5)

Discussion (Line 453 – 466)

 Another importance is that gene silencing of ezrin, but not of radixin and moesin significantly decreased the cell surface expression of P-gp without affecting the levels of the ABCB1 mRNA in LS180 cells cultured at the acidic pH condition which mimics the actual tumor microenvironment. These results are in good agreement with those observed in the normal culture condition. The extracellular acidity is a common hallmark of almost all tumors due to the accumulation of lactic acid and H+, leading to the microenvironmental acidification [28, 68, 69]. Previous studies have shown that microenvironmental acidic pH exerts divergent roles in tumor development such as invasion, dissemination, and metastasis [70-72]. Moreover, a growing body of recent evidence suggests that acquired MDR not only to a conventional chemotherapeutics but also to molecular targeted drugs can also be caused by the highly acidic microenvironment of tumors [69]. Taken together, ezrin may play a key role in the plasma membrane localization of P-gp at the acidic tumor microenvironment, leading to an acquired MDR in a variety of cancer cells.

References (Line 610 – 622, and Line 788 – 806) 

26. Logozzi M, Angelini DF, Iessi E, Mizzoni D, Di Raimo R, Federici C, et al. Increased PSA expression on prostate cancer exosomes in in vitro condition and in cancer patients. Cancer Lett. 2017;403:318-29. Epub 2017/07/12. doi: 10.1016/j.canlet.2017.06.036. PubMed PMID: 28694142.

27. Logozzi M, Mizzoni D, Angelini DF, Di Raimo R, Falchi M, Battistini L, et al. Microenvironmental pH and Exosome Levels Interplay in Human Cancer Cell Lines of Different Histotypes. Cancers (Basel). 2018;10(10). Epub 2018/10/12. doi: 10.3390/cancers10100370. PubMed PMID: 30301144.

28. Boussadia Z, Lamberti J, Mattei F, Pizzi E, Puglisi R, Zanetti C, et al. Acidic microenvironment plays a key role in human melanoma progression through a sustained exosome mediated transfer of clinically relevant metastatic molecules. J Exp Clin Cancer Res. 2018;37(1):245. Epub 2018/10/07. doi: 10.1186/s13046-018-0915-z. PubMed PMID: 30290833.

68. Cairns RA, Harris IS, Mak TW. Regulation of cancer cell metabolism. Nat Rev Cancer. 2011;11(2):85-95. Epub 2011/01/25. doi: 10.1038/nrc2981. PubMed PMID: 21258394.

69. Pillai SR, Damaghi M, Marunaka Y, Spugnini EP, Fais S, Gillies RJ. Causes, consequences, and therapy of tumors acidosis. Cancer Metastasis Rev. 2019;38(1-2):205-22. Epub 2019/03/27. doi: 10.1007/s10555-019-09792-7. PubMed PMID: 30911978.

70. Robey IF, Baggett BK, Kirkpatrick ND, Roe DJ, Dosescu J, Sloane BF, et al. Bicarbonate increases tumor pH and inhibits spontaneous metastases. Cancer Res. 2009;69(6):2260-8. Epub 2009/03/12. doi: 10.1158/0008-5472.CAN-07-5575. PubMed PMID: 19276390.

71. Estrella V, Chen T, Lloyd M, Wojtkowiak J, Cornnell HH, Ibrahim-Hashim A, et al. Acidity generated by the tumor microenvironment drives local invasion. Cancer Res. 2013;73(5):1524-35. Epub 2013/01/05. doi: 10.1158/0008-5472.CAN-12-2796. PubMed PMID: 23288510.

72. Rofstad EK, Mathiesen B, Kindem K, Galappathi K. Acidic extracellular pH promotes experimental metastasis of human melanoma cells in athymic nude mice. Cancer Res. 2006;66(13):6699-707. Epub 2006/07/05. doi: 10.1158/0008-5472.CAN-06-0983. PubMed PMID: 16818644.

Reply comments to #Reviewer 2

We would like to thank #Reviewer 2 for constructive opinion and evaluation on our manuscript. We have carefully read all of your comments and suggestions and have incorporated them into the revised version of manuscript. Detailed responses to your comments are listed below, and we highlighted all changes with word track changes in the file labeled ‘Revised Manuscript with Track Changes’. We hope this revised manuscript would be satisfactory for publication in PLoS One.

Comment 1, line 233, the the

Reply Comments.

As #Reviewer 2 pointed out, we corrected a typo. Thank you for careful reading.

Comment 2, The Figures are in low resolution that were hard to identify.

Reply Comments.

We are sorry it is a bit hard to see the figure in the PDF file because figure resolutions are very low. Therefore, we revised all Figures fallen within the size, dimensions, and resolution using PACE assistance system according to the instruction of this journal as far as possible in the revised version. 

Comment 3, line 301, would it be too high concentration of 50 μM of verapamil?

Reply Comments.

With respect to the concentration of verapamil used in this study, 50 μM of verapamil is often used in the in vitro experiments to inhibit the transport activity of P-gp in many previous studies [1-6]. However, as #Reviewer 2 pointed out, we think that 50 μM of verapamil seems to be a relatively high concentration after careful consideration. Therefore, we performed additional experiments to determine whether lower concentration of verapamil also inhibits the transport activity of P-gp, leading to the accumulation of Rho123, a typical fluorescence substrate for P-gp, in LS180 cells by confocal laser scanning microscopy analysis. Based on the results of additional experiments, we added following sentence and figure in the Results and Supporting Information (S1 Fig. and S1 File Supplemental materials and methods) to the revised version of manuscript.

References for this comment

1. Jodoin J, Demeule M, Beliveau R. Inhibition of the multidrug resistance P-glycoprotein activity by green tea polyphenols. Biochim Biophys Acta. 2002;1542(1-3):149-59. Epub 2002/02/21. doi: 10.1016/s0167-4889(01)00175-6. PubMed PMID: 11853888.

2. Rigalli JP, Ruiz ML, Perdomo VG, Villanueva SS, Mottino AD, Catania VA. Pregnane X receptor mediates the induction of P-glycoprotein by spironolactone in HepG2 cells. Toxicology. 2011;285(1-2):18-24. Epub 2011/04/05. doi: 10.1016/j.tox.2011.03.015. PubMed PMID: 21459122.

3. Shang H, Wang Z, Ma H, Sun Y, Ci X, Gu Y, et al. Influence of verapamil on the pharmacokinetics of rotundic acid in rats and its potential mechanism. Pharm Biol. 2021;59(1):200-8. Epub 2021/02/18. doi: 10.1080/13880209.2021.1871634. PubMed PMID: 33595422.

4. Zhao L, Zhao Y, Schwarz B, Mysliwietz J, Hartig R, Camaj P, et al. Verapamil inhibits tumor progression of chemotherapy-resistant pancreatic cancer side population cells. Int J Oncol. 2016;49(1):99-110. Epub 2016/05/14. doi: 10.3892/ijo.2016.3512. PubMed PMID: 27177126.

5. Zhou Y, Song X, Dong G. Effects of verapamil on the pharmacokinetics of puerarin in rats. Xenobiotica. 2019;49(10):1178-82. Epub 2018/09/04. doi: 10.1080/00498254.2018.1518552. PubMed PMID: 30173622.

6. Zhang Y, Li J, Lei X, Zhang T, Liu G, Yang M, et al. Influence of Verapamil on Pharmacokinetics of Triptolide in Rats. Eur J Drug Metab Pharmacokinet. 2016;41(4):449-56. Epub 2015/04/10. doi: 10.1007/s13318-015-0275-4. PubMed PMID: 25854676.

Results (Line 371 – 373)

Of note, we validated that treatment of LS180 cell with 5, 10, and 50 μM of verapamil dose-dependently increased the uptake capacity of Rho123 (S1 Fig.).

Supporting information (an attached PDF file named S1_fig.)

Supplemental materials and methods

Rho123 accumulation study

 LS180 cells were seeded at a density of 1.0 × 105 cells/dish in polylysine-coated 35-mm glass bottom dish with 14 mm inner diameter (Matsunami Glass, Osaka, Japan) and were incubated overnight at 37°C in a humidified atmosphere with 5% CO2 to allow for attachment. Then cells were pretreated with 5, 10 or 50 μM of verapamil (FUJIFILM Wako Pure Chemical, Osaka, Japan), a classical inhibitor of the P-gp function, for 30 min followed by incubation with 10 μM of Rhodamine123 (Merck, Darmstadt, Germany) as a P-gp substrate in the dark for 2 h at 37°C in a humidified atmosphere with 5% CO2. After washing the cells with D-PBS for three times, photomicrographs were taken at 0.5–1.0-μm intervals for z-axis at an original magnification of 20× with a Nikon A1 confocal laser microscope system (Nikon Instruments, Tokyo, Japan). The two- or three-dimensional images were reconstructed from the obtained pictures using the NIS-Elements Ar Analysis software (Nikon Instruments).

Supplemental figure

S1 Fig. Intracellular Accumulation of Rhodamine123 in LS180 Cells after Treatment with Verapamil in a Concentration-dependent Manner. Cells were treated with verapamil at a concentration of 5, 10, and 50 μM and then incubated for 30 min followed by incubation with 10 μM of Rhodamine (Rho) 123. Representative confocal laser scanning microscopy images of fluorescent Rho123 and of each overlay with phase contrast in LS180 cells. Upper and lower panels indicate Rho123 and merge of Rho123 with phase contrast image, respectively. Scale bars: 50 μm. All the images are representative of at least three independent experiments.

Comment 4, ABCB1 mRNA vs. P-gp mRNA, they are inconsistent in Results and Discussion (line 373)

Reply Comments.

As #Reviewer 2 pointed out, we corrected P-gp mRNA into a suitable expression, i.e., ABCB1 mRNA and used the same terms throughout the document. Thank you for careful reading.

---

## [Decision Letter · Decision Letter 1]

8 Apr 2021

PONE-D-21-04413R1

Subcellular distribution of ezrin/radixin/moesin and their roles in the cell surface localization and transport function of P-glycoprotein in human colon adenocarcinoma LS180 cells

PLOS ONE

Dear Dr. Obata,

Thank you for submitting your manuscript to PLOS ONE. After careful consideration, we feel that it has merit but does not fully meet PLOS ONE’s publication criteria as it currently stands. Therefore, we invite you to submit a revised version of the manuscript that addresses the points raised during the review process.

Your manuscript is essentially acceptable for publication but Reviewer 1 feels that four publications should be cited in your paper.  I would greatly appreciate your consideration of this request. 

We look forward to receiving your revised manuscript.

Kind regards,

Salvatore V Pizzo

Academic Editor

PLOS ONE

Journal Requirements:

Reviewers' comments:

Reviewer's Responses to Questions

**Comments to the Author**

1. If the authors have adequately addressed your comments raised in a previous round of review and you feel that this manuscript is now acceptable for publication, you may indicate that here to bypass the “Comments to the Author” section, enter your conflict of interest statement in the “Confidential to Editor” section, and submit your "Accept" recommendation.

Reviewer #1: All comments have been addressed

Reviewer #2: All comments have been addressed

2. Is the manuscript technically sound, and do the data support the conclusions?

Reviewer #1: Yes

Reviewer #2: Yes

3. Has the statistical analysis been performed appropriately and rigorously? 

Reviewer #1: N/A

Reviewer #2: Yes

4. Have the authors made all data underlying the findings in their manuscript fully available?

Reviewer #1: Yes

Reviewer #2: Yes

5. Is the manuscript presented in an intelligible fashion and written in standard English?

Reviewer #1: Yes

Reviewer #2: Yes

6. Review Comments to the Author

Reviewer #1: The revision has been fairly performed in addressing the raised points. The authors missed to quote some previous papers, that should be included and commented in the revised version, including:

1. Luciani F, Molinari A, Lozupone F, Calcabrini A, Lugini L, Stringaro A, Puddu P, Arancia G, Cianfriglia M, Fais S. P-glycoprotein-actin association through ERM family proteins: A role in P-glycoprotein function in human cells of lymphoid origin. Blood. 2002;99(2):641-648.

2. Fais S. A role for ezrin in a neglected metastatic tumor function. Trends Mol Med. 2004;10(6):249-250.

3. Brambilla D, Fais S. The janus-faced role of ezrin in "linking" cells to either normal or metastatic phenotype. Int J Cancer. 2009;125(10):2239-2245.

4. Federici C, Brambilla D, Lozupone F, Matarrese P, de Milito A, Lugini L, Iessi E, Cecchetti S, Marino M, Perdicchio M, Logozzi M, Spada M, Malorni W, Fais S. Pleiotropic function of ezrin in human metastatic melanomas. Int J Cancer. 2009;124(12):2804-2812.

Reviewer #2: Thank you for the revision which addressed all the issues. Some of the Figures are still in low resolusion, while that may due to the different softwares. It's my opinion that it's now ready for publication.

7. PLOS authors have the option to publish the peer review history of their article (what does this mean?). If published, this will include your full peer review and any attached files.

Reviewer #1: No

Reviewer #2: No

---

## [Author Response · Author response to Decision Letter 1]

8 Apr 2021

Reply comments to #Reviewer 1

We would like to thank #Reviewer 1 for valuable suggestions and suggestions on our revised version of manuscript. We have carefully read your comments and suggestions and have made the corrections in the revised version of manuscript. Detailed responses to your comments are listed below, and we highlighted all changes with word track changes in the file labeled ‘Revised Manuscript with Track Changes’. We hope this revised manuscript would be satisfactory for publication in PLoS One.

Comment. The authors missed to quote some previous papers, that should be included and commented in the revised version, including:

1. Luciani F, Molinari A, Lozupone F, Calcabrini A, Lugini L, Stringaro A, Puddu P, Arancia G, Cianfriglia M, Fais S. P-glycoprotein-actin association through ERM family proteins: A role in P-glycoprotein function in human cells of lymphoid origin. Blood. 2002;99(2):641-648.

2. Fais S. A role for ezrin in a neglected metastatic tumor function. Trends Mol Med. 2004;10(6):249-250.

3. Brambilla D, Fais S. The janus-faced role of ezrin in "linking" cells to either normal or metastatic phenotype. Int J Cancer. 2009;125(10):2239-2245.

4. Federici C, Brambilla D, Lozupone F, Matarrese P, de Milito A, Lugini L, Iessi E, Cecchetti S, Marino M, Perdicchio M, Logozzi M, Spada M, Malorni W, Fais S. Pleiotropic function of ezrin in human metastatic melanomas. Int J Cancer. 2009;124(12):2804-2812.

Reply Comments.

We would like to appreciate #Reviewer 1’s valuable suggestion. As #Reviewer 1 pointed out, we performed additional study to investigate whether ERM proteins also regulate the plasma membrane localization of P-gp under the condition at the acidic pH of tumor microenvironment. Based on the results of additional experiments, we added following sentence in the Introduction, Discussion, and References to the revised version of manuscript.

Introduction (Line 45 – 50)

The members of the ezrin/radixin/moesin (ERM) family of proteins act as linkers between the actin cytoskeleton and various plasma membrane proteins, including P-gp, multidrug resistant protein-2, and epidermal growth factor receptor 2 [19-21]. Several recent studies have demonstrated that ERM play a critical role in the maintenance of cell surface membrane localization and drug efflux function of P-gp in a variety of cancer cells [17, 21-25].

Discussion (Line 467 – 472)

In addition, accumulated evidence suggests that ezrin plays an essential role in both the maintenance of migratory and invasive capacity of tumor cells through the specific interaction with some tumor-associated plasma membrane proteins, which leads to the metastatic behavior of tumor cells [73-75]. Therefore, ezrin may exert multiple function not only in driving the MDR but also in acquiring metastatic phenotype of cancer cells in human tumors.

References (Line 596 – 599, and Line 813 – 822) 

21. Luciani F, Molinari A, Lozupone F, Calcabrini A, Lugini L, Stringaro A, et al. P-glycoprotein-actin association through ERM family proteins: a role in P-glycoprotein function in human cells of lymphoid origin. Blood. 2002;99(2):641-8. Epub 2002/01/10. PubMed PMID: 11781249.

73. Federici C, Brambilla D, Lozupone F, Matarrese P, de Milito A, Lugini L, et al. Pleiotropic function of ezrin in human metastatic melanomas. Int J Cancer. 2009;124(12):2804-12. Epub 2009/02/25. doi: 10.1002/ijc.24255. PubMed PMID: 19235924.

74. Brambilla D, Fais S. The Janus-faced role of ezrin in "linking" cells to either normal or metastatic phenotype. Int J Cancer. 2009;125(10):2239-45. Epub 2009/07/10. doi: 10.1002/ijc.24734. PubMed PMID: 19588507.

75. Fais S. A role for ezrin in a neglected metastatic tumor function. Trends Mol Med. 2004;10(6):249-50. Epub 2004/06/05. doi: 10.1016/j.molmed.2004.04.005. PubMed PMID: 15177187.

Reply comments to #Reviewer 2

We would like to thank #Reviewer 2 for constructive comments through the review process.

---

## [Decision Letter · Decision Letter 2]

16 Apr 2021

Subcellular distribution of ezrin/radixin/moesin and their roles in the cell surface localization and transport function of P-glycoprotein in human colon adenocarcinoma LS180 cells

PONE-D-21-04413R2

Dear Dr. Obata,

We’re pleased to inform you that your manuscript has been judged scientifically suitable for publication and will be formally accepted for publication once it meets all outstanding technical requirements.

Kind regards,

Salvatore V Pizzo

Academic Editor

PLOS ONE

Additional Editor Comments (optional):

Reviewers' comments:

Reviewer's Responses to Questions

**Comments to the Author**

1. If the authors have adequately addressed your comments raised in a previous round of review and you feel that this manuscript is now acceptable for publication, you may indicate that here to bypass the “Comments to the Author” section, enter your conflict of interest statement in the “Confidential to Editor” section, and submit your "Accept" recommendation.

Reviewer #1: All comments have been addressed

2. Is the manuscript technically sound, and do the data support the conclusions?

Reviewer #1: Yes

3. Has the statistical analysis been performed appropriately and rigorously? 

Reviewer #1: N/A

4. Have the authors made all data underlying the findings in their manuscript fully available?

Reviewer #1: Yes

5. Is the manuscript presented in an intelligible fashion and written in standard English?

Reviewer #1: Yes

6. Review Comments to the Author

Reviewer #1: (No Response)

7. PLOS authors have the option to publish the peer review history of their article (what does this mean?). If published, this will include your full peer review and any attached files.

Reviewer #1: No

---

## [Editor Report · Acceptance letter]

30 Apr 2021

PONE-D-21-04413R2 

Subcellular distribution of ezrin/radixin/moesin and their roles in the cell surface localization and transport function of P-glycoprotein in human colon adenocarcinoma LS180 cells 

Dear Dr. Obata:

I'm pleased to inform you that your manuscript has been deemed suitable for publication in PLOS ONE. Congratulations! Your manuscript is now with our production department. 

Kind regards, 

on behalf of

Dr. Salvatore V Pizzo 

Academic Editor

PLOS ONE